# A study on a vehicle semi-active suspension control system based on road elevation identification

Zhengcai Yang[1,2]*, Chuan Shi[1], Yinglin Zheng[2], Shirui Gu[1]

1 School of Automotive Engineering, Hubei University of Automotive Technology, Shiyan, Hubei, China,
2 College of Automotive Engineering, Jilin University, Changchun, Jilin, China

☯ These authors contributed equally to this work.
* yang516516@163.com

**Data Availability Statement:** All relevant data are within the paper and its Supporting Information files.

**Funding:** 1) Funding: This research was funded by the Hubei Provincial Key Research and Development Project, China(No.2020BAB099) 2)

## Abstract

A semi-active suspension system can effectively improve vehicle ride comfort and handling stability, and the active detection of road information is key to achieving semi-active suspension. To improve the road elevation perception ability of vehicles, this study proposes a continuous multiple scanning recursive matching algorithm based on a single-line LIDAR sensor. Radar recursive scanning is used to obtain the multiple superposition data of echo signals, and coordinate matching is realized between historical scanning data and current scanning data. Simultaneously, the sensor height deviation and pitch angle deviation of the sensors are regressed to obtain an accurate pavement elevation. Considering the control effect of the active vehicle suspension, a vehicle suspension model with seven degrees of freedom is established. The semi-active suspension controller is constructed using a diagonal recursive neural network algorithm, and the neural network weight is trained using a genetic algorithm. In addition, a preview diagonal recursive neural network control strategy for semi-active suspension, based on the combination of road elevation information, is proposed. The results of a hardware-in-the-loop co-simulation, which was conducted based on the Simulink control model and dSPACE real-time simulation, revealed that the ride comfort and stability of the vehicle were improved owing to a preview of the elevation information of the road ahead and the active adjustment of the shock absorber of the suspension system.

## 1. Introduction

Suspension systems transfer the force and torque between a vehicle body and the wheels and are used to reduce the impact load from the road surface to the frame. The impact load has an important impact on the vehicle's ride comfort, handling stability, and safety, as well as the service life of tires. Active suspension is an inevitable direction of suspension development [1]. Road roughness is the primary reason for changes in the vehicle body attitude. Therefore, optimization of the suspension strategy of a car can improve the vehicle driving performance to a certain degree; however, the improvement effect is limited because of the hysteresis of the

funder:Zheng-cai YANG,RMB:250000 3) https://kjt.hubei.gov.cn/ 4)The funder plays an important role in the study design, data collection and analysis.

**Competing interests:** The authors have declared that no competing interests exist.

input. Vehicle preview control technology can be used to obtain the road contour and adjust the shock absorber in advance. The ability of a vehicle to obtain the road roughness information of the road ahead in advance and input it to the suspension controller enables the suspension controller to appropriately adjust the damping of the shock absorber according to the road roughness information. When the elevation of the road changes significantly, the vehicle can move along the road with minimum body fluctuation, thus improving vehicle ride comfort.

Current suspension control methods rarely actively detect or perceive road information, and inadequate road information prevents the suspension system from further improving vehicle performance. To address these issues, practical road preview technology has become the research focus of active suspension systems, and accurately obtaining road roughness information is key to achieving the preview effect.

Currently, road height estimation methods are divided into three main categories. The first category includes direct measurement methods, which are simple and effective but have a large workload.

The direct measurement method uses road roughness instruments installed on or connected to the vehicle to measure the road elevation by keeping contact with the ground continually. Xuexun et al. [2] compared and analyzed various pavement roughness measurement methods and concluded that the accuracy of pavement roughness information measured by the direct measurement method is ideal and can reproduce pavement elevation information accurately. However, as a second-order system with a spring and shock absorber, the selection of the spring stiffness and shock absorber damping coefficient will significantly impact the amplitude frequency characteristics of the system, thus affecting the measurement accuracy of the pavement roughness instrument within its resonance frequency range [3]. At present, the road profile [4] and vibration accumulation measuring instrument [5] widely used on vehicles can accurately measure the road elevation information with an amplitude of ±100 mm and a wavelength ranging from 0.5–20 m to 1–50 m; however, the measurement accuracy depends on the vehicle driving speed [6]. In addition, owing to limitations with respect to the structure and installation of these measuring instruments, the vehicle can only be driven at a low speed during the measurement process. Therefore, the direct measurement method is mainly used for the maintenance of the road surface and cannot be used for the on-board real-time measurement of conventional vehicles.

The second category involves the indirect measurement method, which is based on sensor information and requires mature algorithms to process the road signal data. In 2002, Labayrade et al. proposed a real-time nonflat pavement contour detection algorithm [7] that uses the "v-parallax" method to vertically model pavement contours. This method is robust to the contour acquisition of pavements and obstacles. In 2007, Oniga et al. converted three-dimensional (3D) data obtained from a dense stereo parallax map into a rectangular elevation map [8]. The random sample consensus (RANSAC) method was used to fit the quadratic road surface model, and the vertical contour of the road surface and obstacles was then obtained after a transformation, which requires a complex calculation. Researchers at Chang'an University obtained the relevant contours of road objects in advance using image processing techniques, and they performed feature extraction and recognition [9]. The identification data were combined with the acceleration and other sensor data, and transmitted to the control system. In 2014, Shen et al. performed stereo matching on the road ahead in real time to obtain depth information, generate a 3D grid map, and estimate and correct the dynamic pitch angle of the sensor in real time [10]. In 2018, Audi released the 5th generation Audi A8 model that realizes AI active suspension technology through a front R242 camera. These methods, which are based on image processing, require high computational power and a costly controller. The

vision-based detection scheme is sensitive to environmental changes, and its measurement accuracy cannot be ensured in rainy and snowy weather. In addition, the measurement system based on this measurement method cannot provide accurate identification under conditions with potholes, such as gravel roads; hence, this method cannot be used for the continuous measurement of bad roads.

The third category includes an estimation method that is based on the dynamic response of suspension systems. The traditional method [11–15] estimates road conditions using the Kalman filter estimation algorithm or synovial observer [16, 17]. In Reference [18], a method for estimating the road height using the inverse model of the suspension system and the dynamic response of the suspension system was proposed. In Reference [19], a Kalman filter and neural network were jointly used to estimate road height, and its effectiveness depended largely on the quality of data training. The road height estimation method based on dynamic responses is limited by the vehicle state acquisition error, and is only suitable for outputting road grades; this method cannot achieve real-time output. Additionally, some studies adopted the vehicle model and designed the sliding-mode observer to estimate unknown road heights [16, 17]. The road height changes gradually with the system input. This method is only suitable for slow changing pavements, and not for discrete impact pavements such as deceleration belt and pit bag, or long-wave pavements with rapid changes.

Active suspension control algorithms include the proportional integral derivative (PID) control algorithm, optimal control algorithm, robust control algorithm, and neural network control algorithm. In recent years, intelligent control algorithms have combined pavement information with control strategies. For example, the front-road condition and vehicle speed are considered in the suspension control quantity of the preview control algorithm [20]. B Németh et al. combined road information with the robust control algorithm and proposed a fusion strategy of road roughness and driving speed to form the final suspension robust controller [21]. Wang et al. proposed a road surface condition identification approach based on road characteristic value, which can be used in preview control [22].

If the road height information can be obtained in real time, the interference of the road surface on the state estimation of the suspension system can be eliminated [23–25], and the system performance and response speed can be improved through feed-forward compensation of the control system according to the road height information [26]. Therefore, the use of road information in suspension control systems is an area of research interest [27–29] in the field of suspension control.

To obtain the road elevation information in real time, this paper proposes a road elevation recognition method based on a single-line LIDAR. Owing to its structure, LIDAR can be directly installed on existing vehicles. The working principle of transmitting a laser pulse makes the measurement results less affected by speed, the road environment, and light, thereby increasing the measurement accuracy relative to visual schemes. The proposed recursive scanning matching algorithm combines the historical scanning data with the current scanning data to calculate the pavement elevation. Consequently, the elevation information can be identified with a rapid change of pavement impact, such as deceleration belt and discrete impact pavement. In addition, it does not depend on the training accuracy of the deep learning data set, thus yielding a better recognition generalization ability of pavement contour. This paper presents a preview diagonal recursive neural network (Pre-DRNN) control strategy for semi-active suspension based on the real-time acquisition of pavement elevation information. The paper is organized as follows: a recognition method for pavement elevation information is introduced in Section 2, and Section 3 describes the design of the semi-active suspension controller. Then, based on the information presented in Sections 2 and 3, the overall system scheme design is analyzed and discussed in Section 4. The experimental verification is described in Section 5, and conclusions are presented in Section 6.

## 2. Pavement elevation recognition method

LIDAR is a type of radar that functions in the optical band. By sending measurement laser pulses to the target and then receiving the laser signal reflected from the target, the distance information of the target is obtained according to the speed of light and the propagation time between the LIDAR and the target. This study proposes a LIDAR continuous scanning recursive matching algorithm to compute accurate road roughness information based on low-cost single-line LIDAR, and the roughness grade of the pavement is obtained.

When a vehicle is being driven, the vehicle-mounted LIDAR emits laser beams to detect and scan the pavement elevation. The detection area of each beam is fixed based on the installation angle and position of the LIDAR. The detection range of the two moments before and after produces the intersection interval, wherein a data overlap area is created, and the data density in this area is enhanced. When a vehicle travels multiple distances, the data will overlap many times, and the data density will be greatly enhanced. Having a sufficiently large amount of data can significantly compensate for the shortage of single-line LIDAR data. Accordingly, accurate contour information of the front obstacle can be obtained, and measurement errors caused by the road environment and self-motion state can be reduced.

The proposed continuous scanning recursive matching algorithm aims to improve the accuracy of the contour elevation of the obstacle in front of the vehicle by performing multiple continuous scanning recursive matching on the laser pulse echo signal. The specific implementation steps are illustrated in Fig 1.

### 2.1 Calculation of the ground contour elevation

According to the detection range of the vehicle LIDAR, to detect obstacles, the LIDAR is installed in the middle of the front of the vehicle or on both sides of the headlights. The current obstacle elevation is calculated based on the geometric relationship between the laser beam and ground, as shown in Fig 2.

$$n_0 = n_c + n_L + \phi_0, \tag{1}$$

where

$n_c$: Pitch angle offset of LIDAR at installation position;

$n_L$: Relative pitch angle between the vehicle body and wheel;

$\phi_0$: Current LIDAR measures the angle of the beam relative to the sensor housing.

As shown in Fig 2, the scanning angle of the vehicle-mounter LIDAR may vary from approximately 0° to 45° from the horizontal position to the road, and at the two extreme angles, the measuring point on the ground surface is located at an infinite long distance and at the nearest detectable point on the surface. The absolute vertical height from the LIDAR to the ground can be calculated using the installation height of the LIDAR, inclination parameters of the laser beam $z_0$, and horizontal distance from the point to the sensor in the x-axis direction ($x_0$). The calculation formula is as follows:

$$x_0 = d_0 * \cos(n_0), \tag{2}$$

$$z_0 = z - d_0 * \sin(n_0), \tag{3}$$

The vertical distance z from the sensor to the ground is as follows:

$$z = z_{cz} + z_{zd} - x_s * \sin(n_L) + y_s * \sin(w_L), \tag{4}$$

The current scanning data calculates the initial road elevation based on the geometric relationship of the radar to the ground

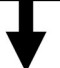

Obtain the distribution probability density of scanned data and achieve quasi-continuous estimation of ground elevation

Recursively call the probability density of historical scanning elevation → Calculate the correlation coefficient between historical scanning and current scanning probability density

Correct the body pitch angle deviation and height deviation to update the elevation of the current scanning data

Historical scanning data → Calculate the final ground contour elevation by fusing the historical scanning data

**Fig 1. Flow chart of algorithm implementation.**

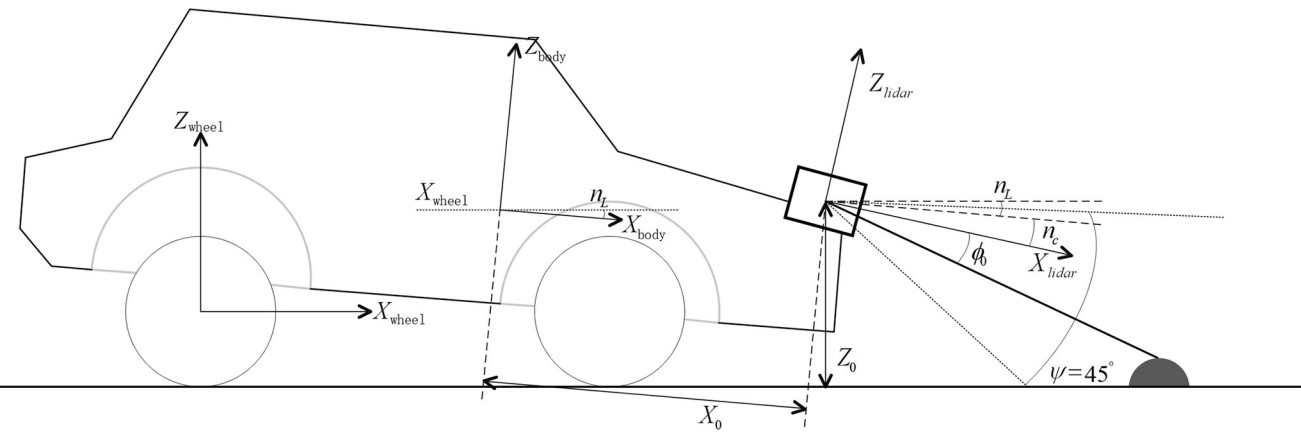

**Fig 2. Geometric relationship between radar and ground.**

Obstacle elevation information can be obtained as:

$$z = z_{cz} + z_{zd} - x_s * \sin(p_L) + y_s * \sin(w_L) - d_0 * \sin(n_c + n_L + \phi_0), \tag{5}$$

where

$z_{cz}$: Installation position offset of vehicle-mounted LIDAR in the vertical direction;

$z_{zd}$: Measurement error caused by the installation of vehicle-mounted LIDAR;

$p_L$: Disturbances caused by vertical bumps and other movements of the vehicle;

$w_L$: Disturbance caused by the left and right movements of the vehicle;

$x_s$: Longitudinal distance between the center of gravity of the vehicle and the LIDAR;

$y_s$: Lateral distance between the center of gravity of the vehicle and the LIDAR;

$d_0$: Distance between LIDAR and measuring point.

## 2.2 Coordinate matching between historical scanning data and current scanning data

According to the above recursive matching algorithm, coordinate matching is performed on the LIDAR scanning data twice through the coordinate transformation relationship, i.e., the polar coordinates are converted into Cartesian coordinates, which can be realized using Eqs (6) and (7):

$$\begin{cases} X_{0,p} = d_{0,p} * \cos(n_{0,p}) \\ Z_{0,p} = z_p - d_{0,p} * \sin(n_{0,p}) \end{cases}, \tag{6}$$

$$\begin{cases} x_{0,n} = d_{0,n} * \cos(n_{0,n}) \\ z_{0,n} = z_n - d_{0,n} * \sin(n_{0,n}) \end{cases}, \tag{7}$$

where

$X_{0,p}$: Distance from a measurement point to the sensor in the x-axis direction obtained from the historical scan;

$z_p$: Vertical distance from the sensor to the ground obtained from the historical scan;

$Z_{0,p}$: Obstacle profile elevation values obtained from the historical scan;

$x_{0,n}$: Distance from a measuring point to the sensor in the X-axis direction obtained from the current scan;

$z_{0,n}$: Obstacle contour height values obtained from the current scan;

$z_n$: Vertical distance from the sensor to the ground obtained by the current scan.

## 2.3 Probability density function

In Section 2.2, the measurement point was considered to be a point, and each distance value measured by LIDAR corresponded to the height value of the obstacle contour. However, in reality, each measurement point is distributed in the form of a spot and not a point. Within a spot, the height value is distributed with a certain probability; this area is called a laser spot. The pulse signal emitted by LIDAR is not evenly distributed in this area, but presents a

Gaussian distribution decreasing from the center to the edge. Therefore, the height value in the measurement point also obeys the Gaussian distribution. Accordingly, a Gaussian distribution probability density function is introduced to represent the probability density of the measurement points:

$$\xi(x) = \frac{1}{\sigma\sqrt{2\pi}} \exp^{(-\frac{1}{2}(\frac{x - x_{ref}}{\sigma})^2)},$$ (8)

where

$x$ is a continuous random variable (in the model, $x$ is the horizontal distance between the LIDAR and measurement point).

$\sigma$ is the standard deviation (or variance)

The algorithm implemented in the above steps is based on the infinitesimal propagation of the measurement points, which is too ideal to truly describe the height profile of the obstacle. If the two scans have the same distance basis, the regression analysis will achieve a superposition of the two scans. For this reason, a coordinate system is established, as shown in Fig 3. The abscissa represents the distance from the measurement point scanned by the LIDAR beam to the LIDAR, and the ordinate represents the height of the obstacle profile. A shift register (which can be understood as an array in the algorithm program) with equidistant sampling points was introduced on the abscissa, and the height value of each scan was inputted using a quantized abscissa value. Thus, the problem of diffusion of the measuring point was solved.

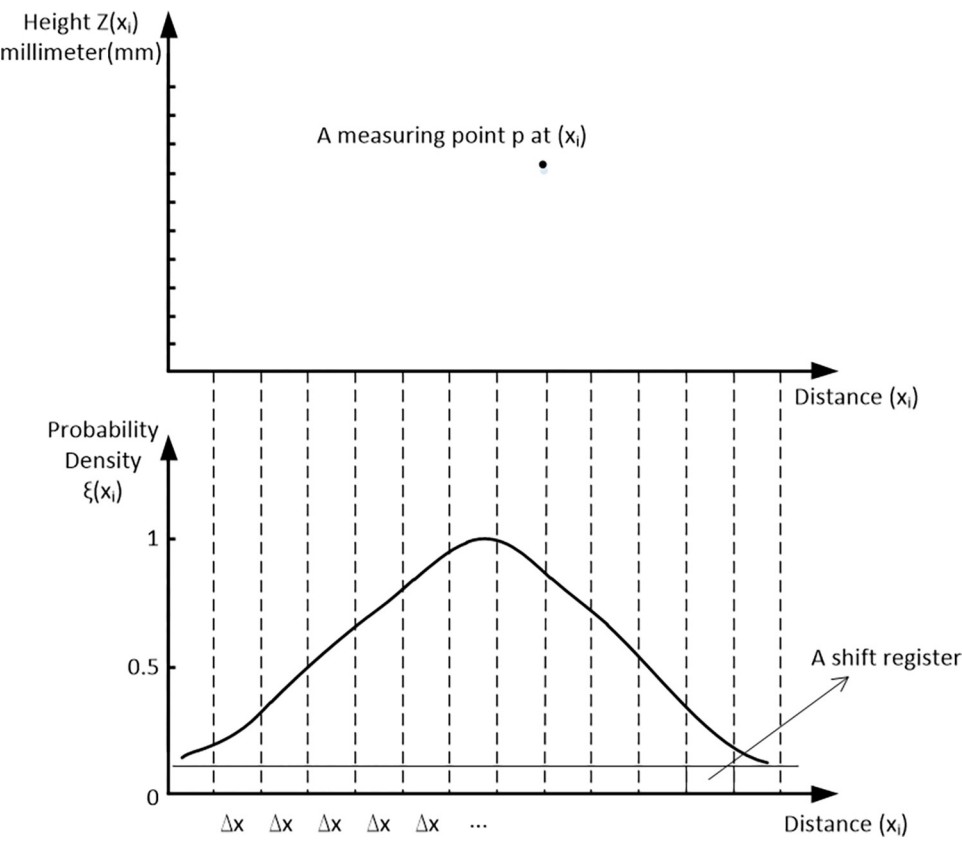

**Fig 3. Application example of shift register with equidistant sampling points.**

Fig 3 shows an example of a shift register application for measurement point p in a scan. A specific abscissa value in the shift register corresponds to a discrete height value, and because the measurement point is shown in the form of a light spot in a real scenario, the discrete measurement value may appear with a certain statistical probability within the light spot range. The elevation value z at point p should consider all possible spots measured at point $x_i$, i.e., the sum of the probability density values of all measurement points at point p should be considered as the comprehensive probability density of point p.

In the shift register, the abscissa is divided at every equidistant sampling point with a distance of $\Delta x$, which is equivalent to rasterizing the distance between the LIDAR scanning point and LIDAR installation position. Fig 3 shows that

$$\begin{aligned} x_{0,cy} &= (x_0, x_0 + 1 * \Delta x_1, \ldots, x_0 + m * \Delta x_1) \\ &= (x_{0,cy,j}, \ldots, x_{0,cy,m}) \\ &\quad j = 0 \ldots .m, x_{0,cy} \in R^{m+1} \end{aligned} \tag{9}$$

The abscissa of the shift register covers the entire measurement range of the LIDAR signal, and a shift register with m + 1 discrete equidistant sampling points can be obtained according to the grid width and maximum scanning distance range. For example, if the measurement range is 0–20 m and the grid width is 10 cm, the shift register has m + 1 = 201 sampling points. In this register, the height value of each measurement point and the probability density distribution are entered via the abscissa value X; Table 1 shows the results.

Assuming that a set of scans has K measurement points and that the different measurement spots were at each sampling point of 0, the probability density of m is

$$\xi_i(x_i) = \frac{1}{\sigma_{cld,j}(x_{0,j})\sqrt{2\pi}} \exp^{\left(-\frac{1}{2}\left(\frac{x_i - x_j}{\sigma_{cld,j}(x_{0,j})}\right)^2\right)} \tag{10}$$

$$i = 1, 2 \ldots k; j = 0, 1 \ldots m$$

The probability density function represents the accuracy of the obstacle height measured in the light spot. The larger the peak value of the probability density, the more concentrated the probability distribution and the higher the accuracy of the measurement. Using the probability density function, the measured data can be processed continuously to obtain a dense obstacle profile height curve.

## 2.4 Quasi-continuous estimation of the obstacle profile

Each time a new scan is generated, K height values that are represented in discrete form in terms of distance are obtained. However, in practice, there is a corresponding height value at each position of the shift register where the probability is nonzero. Assuming that the current scan is made up of K measurement points, the current estimated value of the height value can be calculated using m + 1 discrete grid points in the shift register. Then, according to the product of the probability density matrix and the vector of K height values (taking the sum of n probability density functions of one scan as a unified standard).

**Table 1. Registers with equidistant sampling points for saving scanned data.**

| $x_{0,cy},1$ | $x_{0,cy},2$ | . | $x_{0,cy},m$ |
|---|---|---|---|
| $z_{0,cy},1$ | $z_{0,cy},2$ | . | $z_{0,cy},m$ |
| $\xi_0,1$ | $\xi_0,2$ | . | $\xi_0,m$ |

The probability density value at the first sampling point is $\xi_{0,1}, \xi_{0,2}\ldots\xi_{0,k}$. The estimated value of the corresponding height value can be calculated using normalization:

$$z_{0,cy} = \sum \left( z_{0,cy} = \frac{\sum(\xi_0 * z_{0,i})}{\sum \xi_0} \right) \quad i = 1, 2 \ldots k, \tag{11}$$

where the weighted sum of the scanning data is:

$$\sum \xi_0 = \xi_{0,1} + \ldots + \xi_{0,k}, \tag{12}$$

The quasi-continuous estimation of the probability density values and the corresponding height values of the sampling points can also be obtained by standardization.

The value of the probability density at the mth sampling point is $\xi_{m,1}, \xi_{m,2}\ldots\xi_{m,k}$. The estimated value of the corresponding height value can be calculated by normalization:

$$z_{m,cy} = \frac{\sum(\xi_m * z_{m,i})}{\sum \xi_m}, \tag{13}$$

where the weighted sum of the scanning data is

$$\sum \xi_m = \xi_{m,1} + \ldots + \xi_{m,k}, \tag{14}$$

According to the above algorithm, a quasi-continuous estimation of the obstacle contour of each equidistant point in each scan can be obtained.

## 2.5 Obtaining an accurate obstacle profile height

An algorithm that uses the data of all the current and historical scans can greatly improve the signal quality of the obstacle. This goal can be achieved by recursively calling the scan-matching algorithm between the historical and current scans at each scan. The recursive superposition algorithm is briefly described by the following formula.

Recursive call for current scan:

$$\left( z_{0,cy,n}, \sum \xi_{0,n} \right) = f(x_{0,cy,n}, t), \tag{15}$$

$$\left( z_{0,cy,p}, \sum \xi_{0,p} \right) = f(x_{0,cy,p}, t), \tag{16}$$

where

$z_{0,cy,n}$: Calculated height in the current scan;

$\Sigma\xi_{0,n}$: Sum of the probability density functions of the first sample point in the current scan;

$z_{0,cy,p}$: Calculated value of height in the historical scan;

$\Sigma\xi_{0,p}$: Sum of the probability density functions of the first sampling point in the historical scan;

$x_{0,cy,n}$: Distance from the point to the sensor in the x-axis direction in the current scan;

$x_{0,cy,p}$: Distance from the point to the sensor in the x-axis direction in the history scan.

## 2.6 Calculation of the altitude value deviation and pitch angle deviation

In the recursive superposition algorithm, we should also consider the calculations of related variables. The error of the road contour height value should be considered when the current

(new) scan and historical (old) scan are overlapped by the regression method. The height shift or height error in the shift register can be expressed as follows:

$$\varepsilon z_{0,cy}(x_{0,cy}) = z_{0,cy,p} - z_{0,cy,n}, \tag{17}$$

where

$z_{0,cy,p}$: Obstacle profile height value of the history scan;

$z_{0,cy,n}$: Obstacle profile height value of the current scan;

$\varepsilon z_{0,cy}$: Height shift or height error in the shift register.

To overlap the new scan and old scan by linear regression, it is necessary to determine the weight with which to consider the height error of the obstacle contour corresponding to each abscissa value in the shift register. In simple terms, the error of the height value needs to be considered only at locations with a high normalized probability density of the current and historical scans. Therefore, regression analysis is considered only within the minimum intersection of the probability density distributions of the two scans because only the overlap of the two scans is relevant. Considering this, a correlation coefficient R is introduced, which can be calculated from the minimization criterion of the generalized probability density function as follows:

$$R = \|\sum \xi_{0,n}, \sum \xi_{0,p}\|_{min}, \tag{18}$$

where

$\Sigma\xi_{0,n}$: Probability density function in the current scan;

$\Sigma\xi_{0,p}$: Sum of the probability density functions in the historical scan.

A parameter, i.e., the correlation coefficient R, is added to the recursive superposition algorithm, which indicates that factors such as the light spot plane distribution of the laser measuring points and the subsequent probability density distribution of the height values corresponding to the measuring points are considered when determining the height value deviation and pitch angle deviation of the obstacle contour of the current and historical scans. The following new relationship can be derived between the current and historical scan data:

$$\left(R, R*x_{0,cy}\right) * \begin{pmatrix} \Delta z \\ \Delta n \end{pmatrix} = R * \varepsilon z_{0,cy}, \tag{19}$$

where

$x_{0,cy}$: Distance of the points from the LIDAR in the x-axis direction in a shift register with equidistant sampling points;

$\Delta n$: Pitch angle deviation between the old and new scans;

$\Delta z$: Height value deviation between the old and new scans.

This equation is an overdetermined system of equations similar to Ax = B. The above equation can be solved using linear regression. The generalized inverse matrix A + of matrix A is

constructed as follows:

$$\hat{x} = A^+ * b = (A^T * A)^{-1} * A^T * b$$

$$\begin{pmatrix} \Delta z \\ \Delta n \end{pmatrix} = \left( \left( R, Rx_{0,cy} \right)^T \cdot \left( R, Rx_{0,cy} \right) \right)^{-1} \cdot \left( R, Rx_{0,cy} \right)^T \cdot \left( R, \varepsilon z_{0,cy} \right), \tag{20}$$

According to the above formula, the height deviation and pitch angle deviation $\Delta n$ for multiple groups of obstacle contours can be obtained. The least square method can be used to determine the optimal value of the height deviation $\Delta z$ and the optimum value of the pitch angle deviation $\Delta n$. The height correction value of the new scan data, $z_{0,cy,xz}$, is calculated as follows:

$$z_{0,cy,xz} = z_{0,cy,n} + \Delta n * x_{0,cy} + \Delta z, \tag{21}$$

where the old and new scans are superimposed.

## 2.7 Fusion of the old and new scan data

Through an implementation of the above steps, the current and historical scan data can be combined using the correction obtained by the previous recursive superposition coincidence. When the data of a new scan are added to the saved data of a historical scan, the summary probability density of all previous scans increases the probability density of the new scan as follows:

$$\sum \xi_{0,sum} = \sum \xi_{0,n} + \sum \xi_{0,p}, \tag{22}$$

where

$\Sigma \xi_{0,sum}$: Summary probability density of all scans included at the first sample point;

$\Sigma \xi_{0,p}$: Sum of probability densities of historical sweeps;

$\Sigma \xi_{0,n}$: Sum of probability densities for the current scan.

Under the premise of considering the new probability density, the updated average height of the road contour can be calculated as follows:

$$z_{0,cy,sum} = \frac{z_{0,cy,p} * \sum \xi_{0,p} + z_{0,cy,xz} * \sum \xi_{0,n}}{\sum \xi_{0,sum}}, \tag{23}$$

This average height value $z_{0,cy,sum}$ is the final accurate obstacle contour height value. An accurate obstacle contour in front of the vehicle can be obtained by first replacing the height value of the obstacle contour of the current scan with $z_{0,cy,sum}$, followed by performing recursive superposition of the current scan and the new scan, calculating the height value of the next scan, and then repeatedly performing recursive superposition.

## 3. Design of semi-active suspension controller

### 3.1 Dynamic model of semi-active suspension system

An automobile suspension system is a multi-degree-of-freedoms (DOFs) multi-body system, and the kinematic and mechanical relations between the components are very complex. Therefore, it is difficult to use traditional calculation methods to analyze its kinematics and dynamic

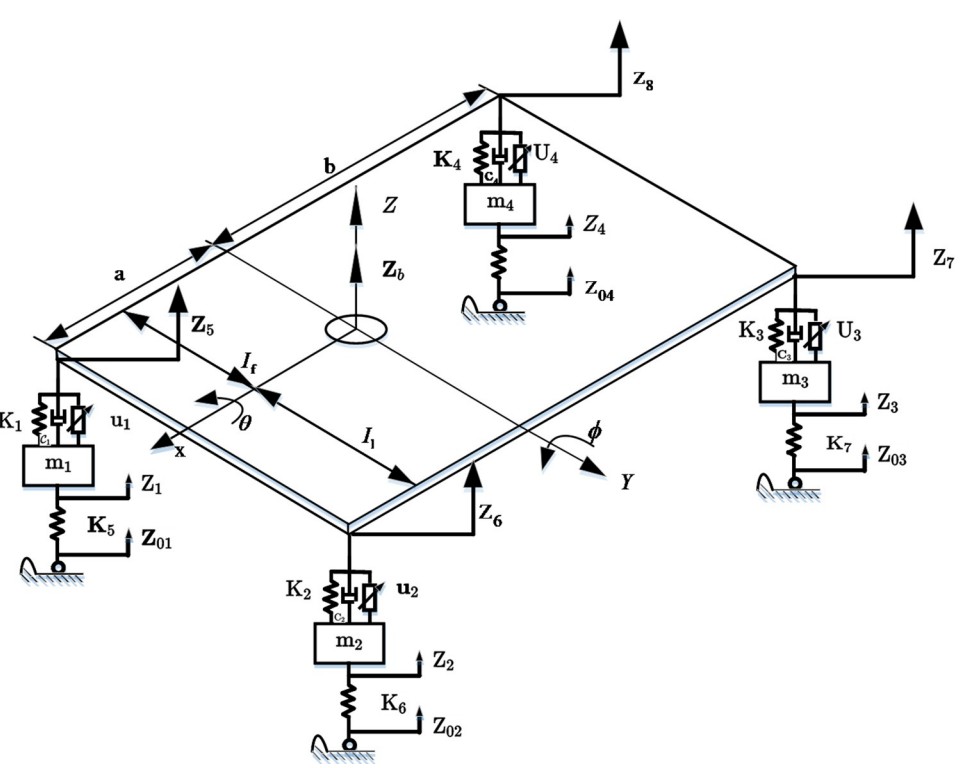

**Fig 4. Dynamic model of active suspension.**

characteristics. Modeling and simulation analysis are economical and efficient methods for studying the control effect of suspension systems.

Assuming that the vehicle body is rigid and the suspension motion has three DOFs, namely vertical vibration, pitch, and roll, and four DOFs of vertical motion of four wheels, a seven DOFs vehicle model of the entire vehicle is established to fully reflect the problems of vertical jump, pitch change, and roll. Among them, the model mechanical (stiffness, damping) and mass (mass, moment of inertia) parameters are derived from real vehicle data, as shown in Fig 4, and the semi-active suspension model parameters are listed in Table 2.

**Table 2. Significance of the semi-active suspension model parameters.**

| Parameter | Definition |
|---|---|
| $z_b$ | Vertical displacement of body center of mass |
| $\phi$ | Pitch angle displacement |
| $\theta$ | Roll angle displacement |
| $a, b$ | Distance between the center of mass and the front and rear axis |
| $l_b, l_r$ | Distance of the center of mass from the left and right wheel |
| $u_1, u_2, u_3, u_4$ | Adjust the input of control quantity |
| $z_1, z_2, z_3, z_4$ | Vertical vibration displacement of unsprung mass |
| $z_5, z_6, z_7, z_8$ | Auxiliary displacement |
| $z_{01}, z_{02}, z_{03}, z_{04}$ | Pavement excitation input |
| $k_1, k_2, k_3, k_4$ | Equivalent stiffness of suspension spring |
| $k_5, k_6, k_7, k_8$ | Tire dynamic stiffness |
| $c_1, c_2, c_3, c_4$ | Equivalent damping of shock absorber |

For the dynamic analysis of the suspension, the acceleration of the suspension center of mass can be expressed as follows:

$$m_c\ddot{z}_b = k_2(z_2 - z_6) + k_3(z_3 - z_7) + k_4(z_4 - z_8)+$$

$$k_1(z_1 - z_5) + c_2(\dot{z}_2 - \dot{z}_6) + c_3(\dot{z}_3 - \dot{z}_7) + c_4(\dot{z}_4 - \dot{z}_8) +$$

$$c_1(\dot{z}_1 - \dot{z}_5) - u_2 - u_3 - u_4 - u_1 + m_c g, \tag{24}$$

The suspension pitch angular velocity can be expressed as follows:

$$J_p\ddot{\phi} = -[k_2(z_2 - z_6) + c_2(\dot{z}_2 - \dot{z}_6) + k_1(z_1 - z_5) + c_1(\dot{z}_1 - \dot{z}_5)]a+$$

$$[k_4(z_4 - z_8) + c_4(\dot{z}_4 - \dot{z}_8) + k_3(z_3 - z_7) +$$

$$c_3(\dot{z}_3 - \dot{z}_7)]b - (u_1 + u_2)a + (u_3 + u_4)b, \tag{25}$$

The suspension roll angle acceleration can be expressed as follows:

$$J_p\ddot{\phi} = -[k_4(z_4 - z_8) + c_4(\dot{z}_4 - \dot{z}_8) + k_1(z_1 - z_5) + c_1(\dot{z}_1 - \dot{z}_5)l_r+$$

$$k_2(z_2 - z_6) + c_2(\dot{z}_2 - \dot{z}_6) + k_3(z_3 - z_7) +$$

$$c_3(\dot{z}_3 - \dot{z}_7)]l_l - (u_1 + u_4)l_r + (u_2 + u_3)l_l, \tag{26}$$

The established seven DOF parameters are $z_b$, $\varphi$, $\theta$, $z_1$, $z_2$, $z_3$ and $z_4$. The 4-wheel acceleration can be expressed as follows:

$$m_1\ddot{z}_1 = k_5(z_{01} - z_1) + k_1(z_5 - z_1) + c_1(\dot{z}_5 - \dot{z}_1) + u_1 + m_1 g, \tag{27}$$

$$m_2\ddot{z}_2 = k_6(z_{02} - z_2) + k_2(z_6 - z_2) + c_2(\dot{z}_6 - \dot{z}_2) + u_2 + m_2 g, \tag{28}$$

$$m_3\ddot{z}_3 = k_7(z_{03} - z_3) + k_3(z_7 - z_3) + c_3(\dot{z}_7 - \dot{z}_3) + u_3 + m_3 g, \tag{29}$$

$$m_4\ddot{z}_4 = k_8(z_{04} - z_4) + k_4(z_8 - z_4) + c_4(\dot{z}_8 - \dot{z}_4) + u_4 + m_4 g, \tag{30}$$

The establishment of a road excitation input model is the basis for studying the dynamic response and control of a semi-active suspension. In this study, the intermittent bumpy road excitation was used as the road disturbance input. The road surface input is described by a time-domain expression of the filtered white noise:

$$\dot{z}_{01} = -2\pi n_0 z_{01} + 2\pi\sqrt{G_0 v w_1}, \tag{31}$$

$$\dot{z}_{02} = -2\pi n_0 z_{02} + 2\pi\sqrt{G_0 v w_2}, \tag{32}$$

$$\dot{z}_{03} = -2\pi n_0 z_{03} + 2\pi\sqrt{G_0 v w_3}, \tag{33}$$

$$\dot{z}_{04} = -2\pi n_0 z_{04} + 2\pi\sqrt{G_0 v w_4}, \tag{34}$$

where $n_0$ is the lower cut-off frequency, $n_0 = 0.01 Hz$; $G_0$ is the pavement roughness coefficient,

$w_i$ is Gaussian white noise with mean = 0 and intensity = 1, and $v$ is the forward speed of the vehicle.

## 3.2 Design of diagonal recurrent neural network controller

A diagonal recurrent neural network (DRNN) is a simplified and fully connected neural network, wherein no information is exchanged among the units in the hidden layer, which significantly simplifies the model and ensures learning speed and the model suitability for the control requirements of semi-active suspension. In this paper, a DRNN intelligent control algorithm is proposed for semi-active suspension control.

The diagonal RNN includes an input layer, a hidden layer, and an output layer. The number of input and output layers can be adjusted according to the number of inputs and outputs. The hidden layer is the intermediate layer, and its control number is determined by the number of input and output layers. The structure of the DRNN model is shown in Fig 5.

In this study, the vehicle vertical acceleration, suspension dynamic stroke, and tire dynamic stroke related to vehicle ride comfort and response-handling stability were selected as the inputs of the neural network algorithm. The layers of the neural network used in this study were as follows:

(1) The first layer was the input layer, which had n input nodes. Its input quantity $x_i(k)$ included the following:

Suspension vertical acceleration:

$$I_1(k) = [\ddot{z}_b], \tag{35}$$

Suspension dynamic stroke:

$$I_2(k) = [z_2 - z_6, z_3 - z_7, z_4 - z_8, z_1 - z_5], \tag{36}$$

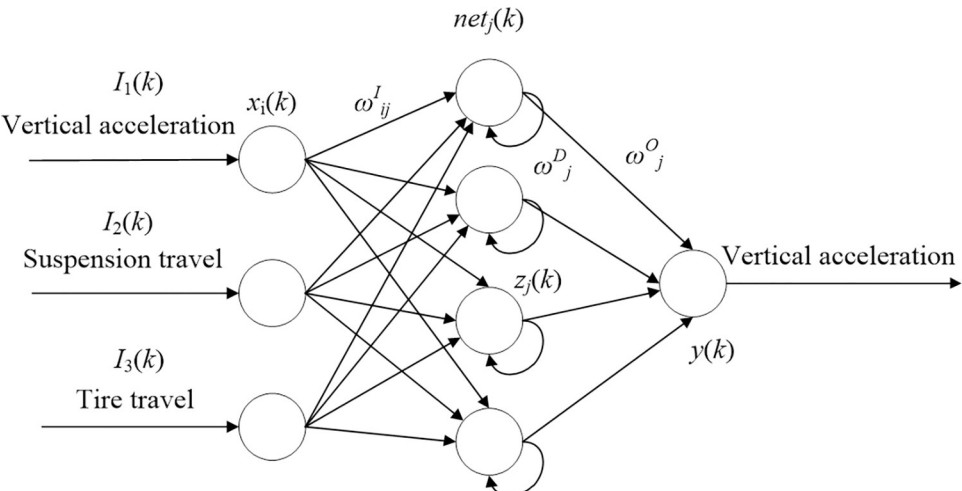

**Fig 5. Schematic diagram of diagonal recurrent neural network model.**

Tire dynamic stroke:

$$I_3(k) = [z_{01} - z_1, z_{02} - z_2, z_{03} - z_3, z_{04} - z_4], \tag{37}$$

(2) The second layer was the hidden layer, and the input was

$$net_{j1}(k) = \omega_j^D z_j(k-1) + \sum_{i=1}^{3} \omega_{ij}^I I_i(k), \tag{38}$$

where $\omega_j^D$, $\omega_{ij}^O$ are the weights of the input and hidden layers, respectively.

$$z_j(k) = f(net_j(k)) = \frac{1 - \exp(-net_j(k))}{1 + \exp(-net_j(k))}, \tag{39}$$

Here, $z_j(k)$ takes the Sigmoid function as the activation function of the hidden layer.

(3) The third layer was the output layer, and the output quantity was

$$y(k) = \sum_{j=1}^{m} \omega_j^O z_j(k), \tag{40}$$

where $\omega_j^O$ is the weight of the output layer.

(4) Assuming that the target value of the vehicle suspension control system was $y_d(k)$, the energy error function was obtained as

$$E(k) = \frac{1}{2}(y_d(k) - y(k))^2, \tag{41}$$

(5) For the recursive layer:

$$\Delta\omega_j^D = -\eta_i^D \frac{\partial E(k)}{\partial \omega_j^D} = -\eta_{ij} \frac{\partial E(k)}{\partial net_j(k)} \frac{\partial net_j(k)}{\partial \omega_j^D}, \tag{42}$$

where $-\eta_i^D$ is the learning rate of the recursive layer; thus, the new weight of the recursive layer can be obtained by $\omega_j^D(k+1) = \omega_j^D(k) + \Delta\omega_j^D(k)$.

(6) Input and hidden layers.

$$\Delta\omega_{ij}^I(k) = -\eta_{ij} \frac{\partial E(k)}{\partial \omega_{ij}^I} = -\eta_{ij} \frac{\partial E(k)}{\partial \omega_j^O} \frac{\partial \omega_j^O}{\partial \omega_{ij}^I}, \tag{43}$$

where $\eta_{ij}$ is the learning rate between the input and the hidden layers. The input layer to the hidden layer can be adjusted according to $\omega_{ij}^I(k+1) = \omega_{ij}^I(k) + \Delta\omega_{ij}^I(k)$

It is important to train the connection weight value between each layer of the neural network system. In this study, genetic algorithms were used for neural network weight training, which proceeded as follows:

1. First, the weights were coded accordingly, and several weights of the entire network were coded using binary coding. The neural network in this study comprised three input nodes, namely $I_1(k)$, $I_2(k)$, and $I_3(k)$; the number of intermediate nodes was $net_j(k)$; and the number of output nodes was 1, which was the vertical acceleration.

2. The individual network weights obtained by the previous encoding step were trained, and the optimal solution of the network weights was obtained through the decoding function of the individual weights. When calculating the fitness of the weights between the layers, the weights were evaluated through the performance index of the semi-active suspension control system, such as the K & C characteristics of the suspension system. Then, the corresponding adaptation value was obtained, and the output of the network was calculated. When an individual search is performed using a genetic algorithm, the optimal algorithm can be adopted to obtain the optimal value of the network weight.

## 4. Overall system scheme design

The DRNN controller designed in Section 3 has self-organization and self-learning functions. It can satisfactorily perform fault tolerance, nonlinear approximation, and adaptive control; however, the feedback control of the hydraulic actuator is effective only after the suspension is excited and the state changes. Therefore, to achieve a more effective control effect, a Pre-DRNN fusion algorithm was developed, and the road information obtained in advance was taken as the feedforward term and combined with the semi-active suspension feedback control.

As shown in Fig 2, the set distance of the road preview is $d_0{}^*\cos n_0$, where V is the speed of the vehicle. The time between the preview point and front wheel was defined as $t_{pre} = (d_0{}^*\cos n_0)/V$. When feedforward control was added, it was necessary to judge the elevation change of the road ahead and to make further adjustments after the change in the road roughness reached the threshold. According to the difference in pavement elevation change and change rate, the control strategy can output different control forces according to specific road conditions.

The known quantities in preview control include the following items: preview time $t_{pre}$, road input z(t) at the current time, the road input of the preview point z(t + $t_{pre}$), and the rate of change of the road roughness $\dot{z}(t)$. Combined with the road roughness information obtained by the continuous scanning recursive matching algorithm of a single-line LIDAR sensor, the rate of change of the road roughness can be taken as an additional criterion, and the DRNN damping control response can be advanced to the time before the arrival of road impact. At this time, if the damping change is appropriate, the displacement and acceleration of the vehicle body can be absorbed by the action of the shock absorber to a great extent. Therefore, a pre-aiming point can be set at $t_{pre}{-}\Delta t$ from the current wheel position to assess the rate of change of road roughness. First, let $\Delta h$ = z(t)–z(t–$\Delta t$), where $\Delta h$ represents the difference between the road roughness at the preview point and the road roughness at the sub-preview point. The international standard ISO/ TC108/SC2N-67 reflects the road roughness using the road power spectral density. To describe the change in road elevation with time, this study converted the road spatial power spectral density into time power spectral density to meet the analysis requirements of the time-domain response of the suspension system. The elevation change threshold and damping adjustment value are controlled by the subsections, and the relationship between the road elevation change and surface roughness is established, as shown in Fig 6. The road roughness is divided into six levels: A, B, C, D, E, and F.

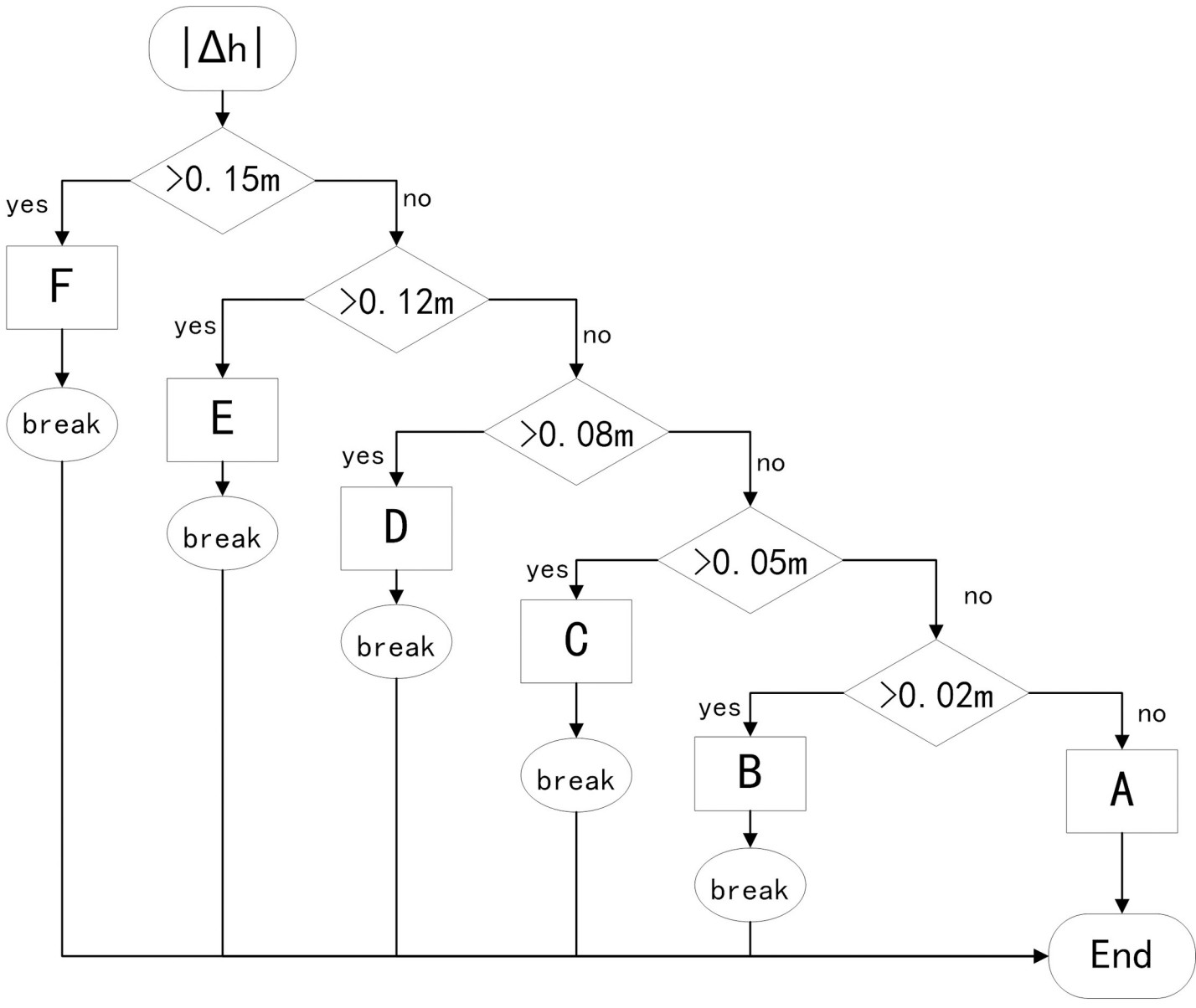

**Fig 6. Corresponding relationship between road elevation and surface roughness levels.**

According to the relationship with the vehicle structure, the vertical displacement of the vehicle body caused by the road input is affected by the shrinkage of the shock absorber to a certain extent; however, it is not always the case that the contraction of the shock absorber weakens the vertical displacement of the body. When the vehicle body is simulated by a convex road surface, the reduction in the damping of the shock absorber will increase the contraction degree of the shock absorber, i.e., the dynamic stroke of the suspension will increase. At this time, the large upward displacement of the original vehicle body is transformed into a dynamic stroke of the suspension. However, when the vehicle body is simulated by the input of the concave road surface, because of the installation position of the shock absorber, the downward displacement of the vehicle body is the contraction displacement of the shock absorber, which can also be roughly equal to the dynamic stroke of the suspension. Therefore, to slow down

**Preview diagonal recursive neural network(Pre-DRNN) fusion algorithm**

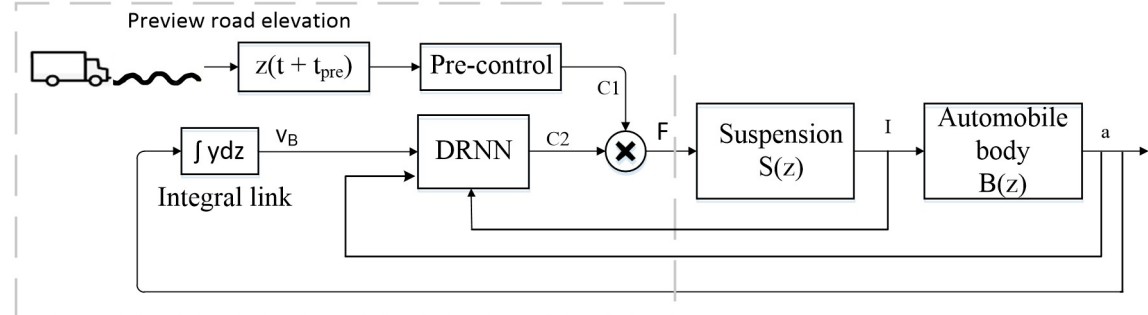

**Fig 7. Block diagram of semi-active suspension control strategy with preview function.**

the vertical displacement of the vehicle body, it is necessary to increase the damping of the shock absorber.

As shown in Fig 7, the semi-active suspension control strategy with the preview function is as follows:

1. The current road roughness is used as an input to stimulate the suspension system S(z) to produce a change in the vertical speed. The vertical displacement output of the wheel is I.

2. The damping damper changes the vertical acceleration a of the body system B(z). The output of the body's vertical speed is $v_B$.

3. The automobile body vertical acceleration a and suspension travel I are input into the suspension controller DRNN, and the system determines whether there are changes in the

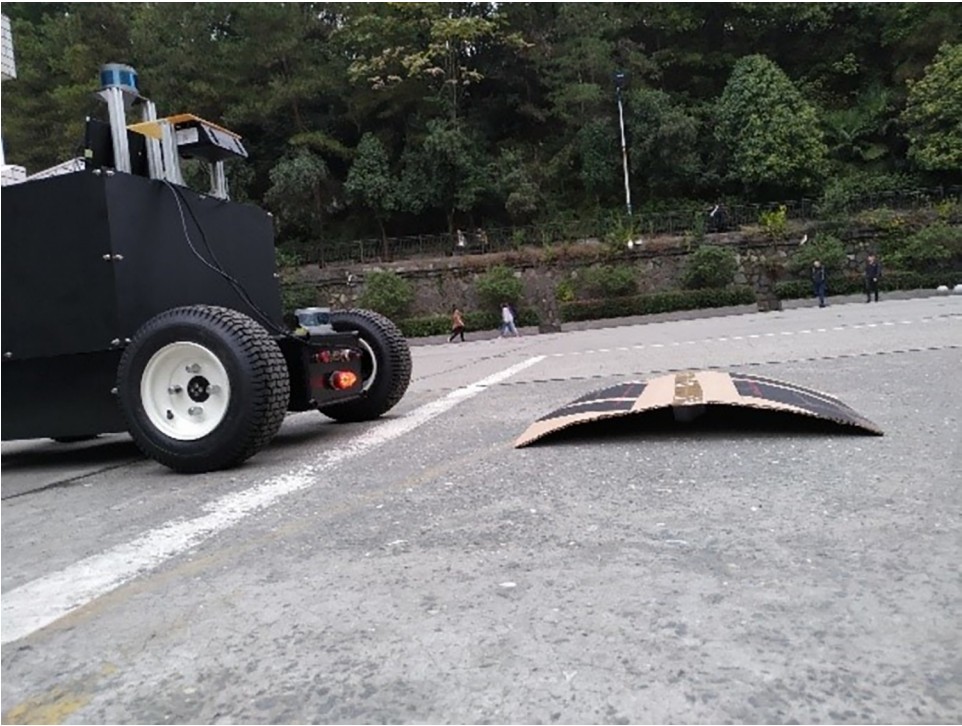

**Fig 8. LIDAR detection of obstacle settings.**

current output hydraulic cylinder damping value C2, which affects the force F and is transmitted to the vehicle body. Simultaneously, the DRNN controller optimizes the output of the next step based on the current feedback information.

4. After adding preview control Pre-control, the system can obtain information about the road ahead at an interval $t_{pre}$ in advance and assess whether the rate of change of the road roughness reaches the threshold. If it reaches a certain threshold, the preview system delays the $t_{pre}$ time to output a damping value C1 and jointly controls the shock absorber with the damping value C2 output by the DRNN controller to optimize the output target of the control quantity.

## 5. Experimental demonstration

### 5.1 Experimental demonstration of the recursive matching algorithm

In this study, the above algorithms were tested by comparing the accuracy of the front obstacle contour at different distances to verify whether the obtained obstacle contour gradually converges to the real contour with an increase in the number of iterations of the algorithm. The LIDAR laser was built on the chassis of a small unmanned vehicle, and the installation position was determined. An arched object with a length of 60 cm, width of 38 cm, and maximum

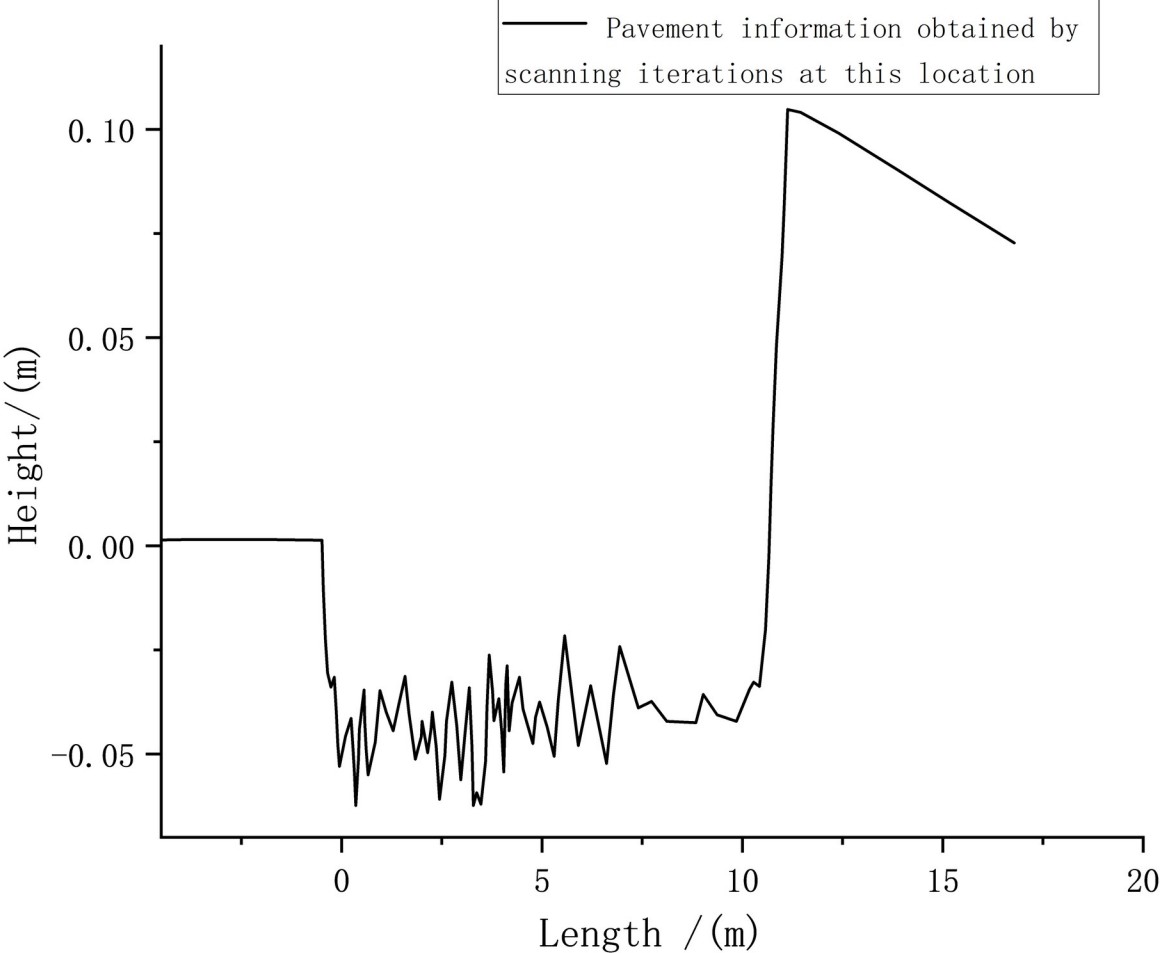

**Fig 9. Road surface information obtained by scanning iterations at the starting point.**

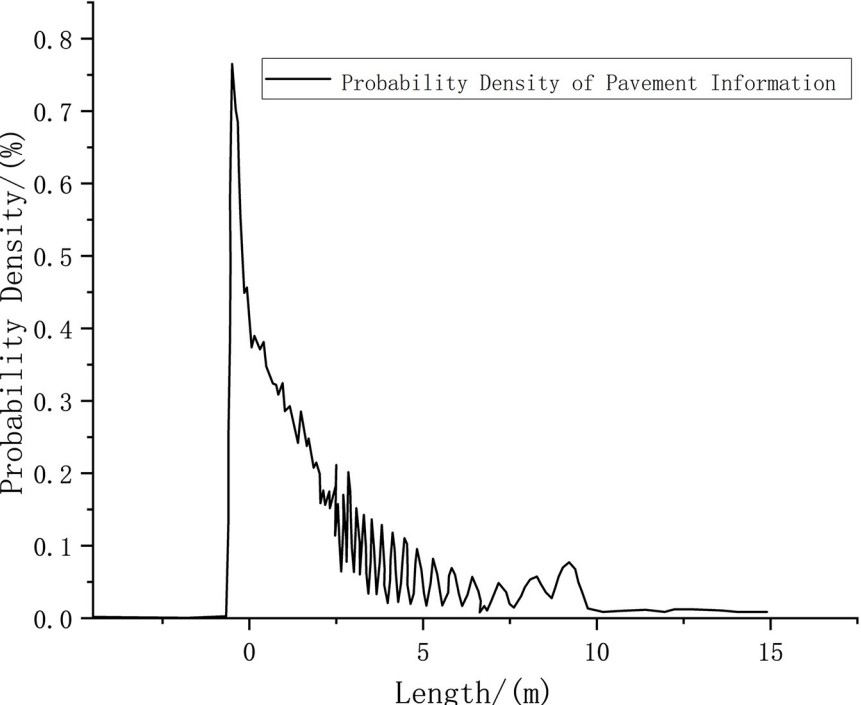

**Fig 10. Obstacle information at the starting point.**

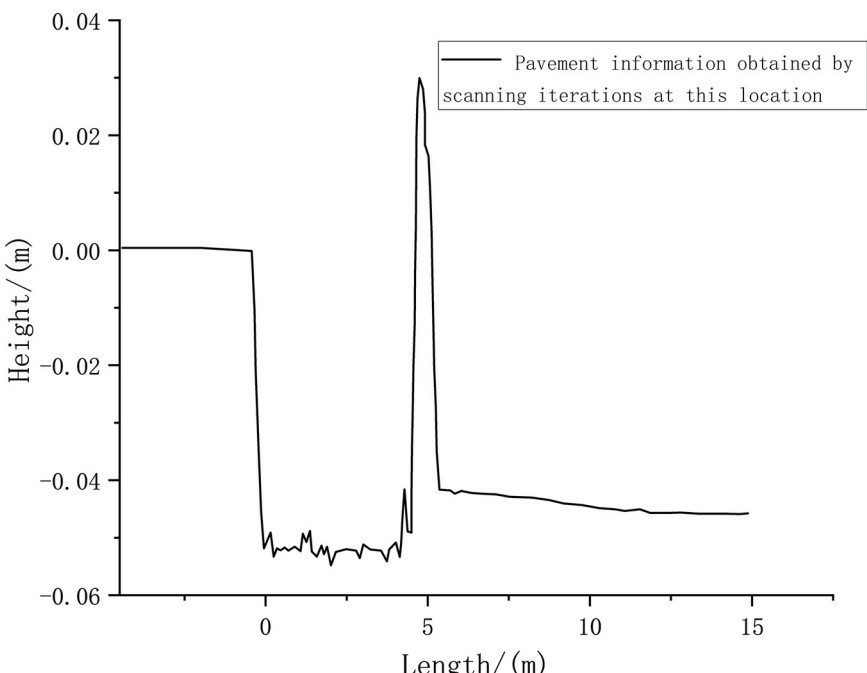

**Fig 11. Scanning iterative road surface information after vehicle travel for 5 m.**

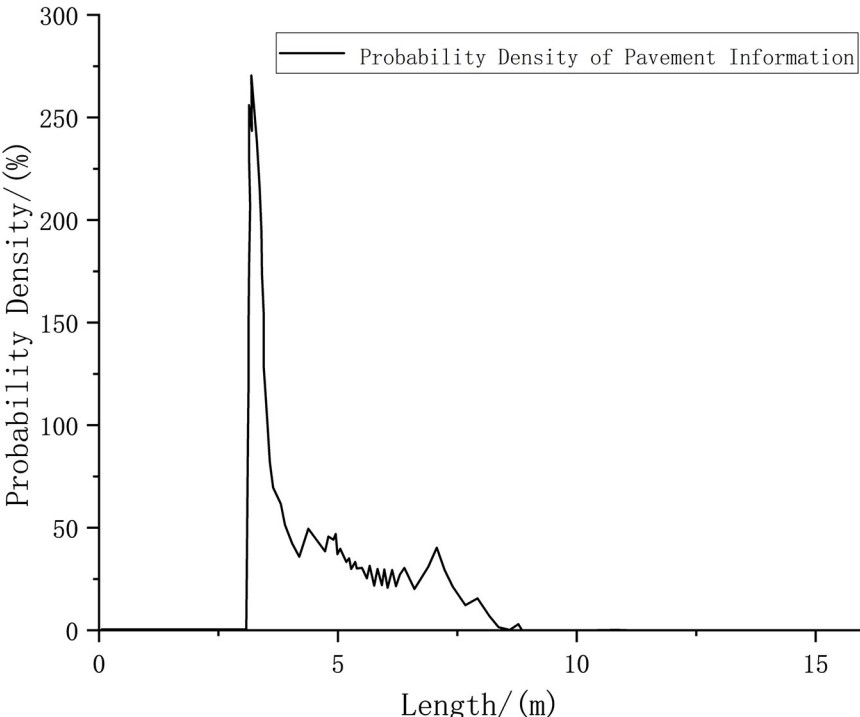

**Fig 12. Road surface probability density for a vehicle driving 5 m.**

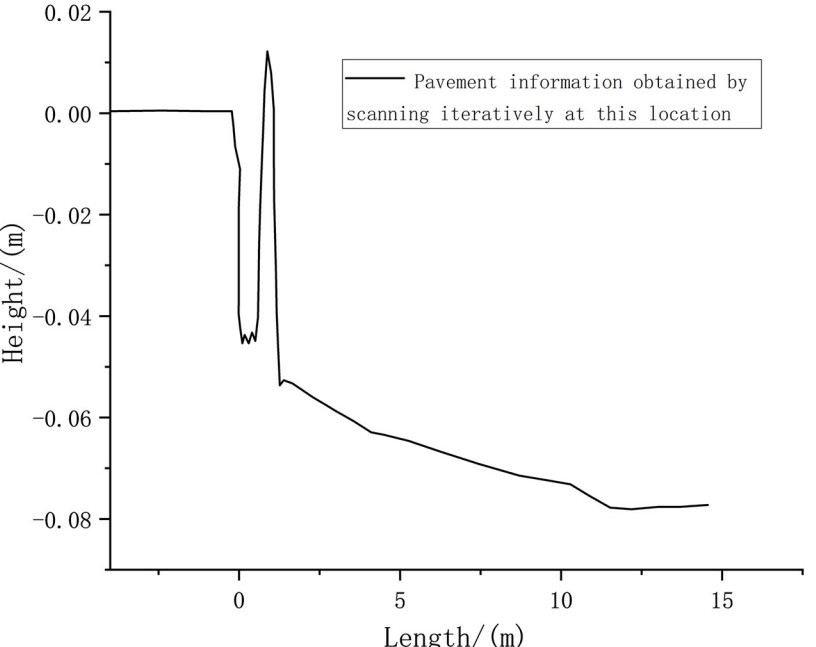

**Fig 13. Scanning iterative pavement information for a vehicle traveling 9 m.**

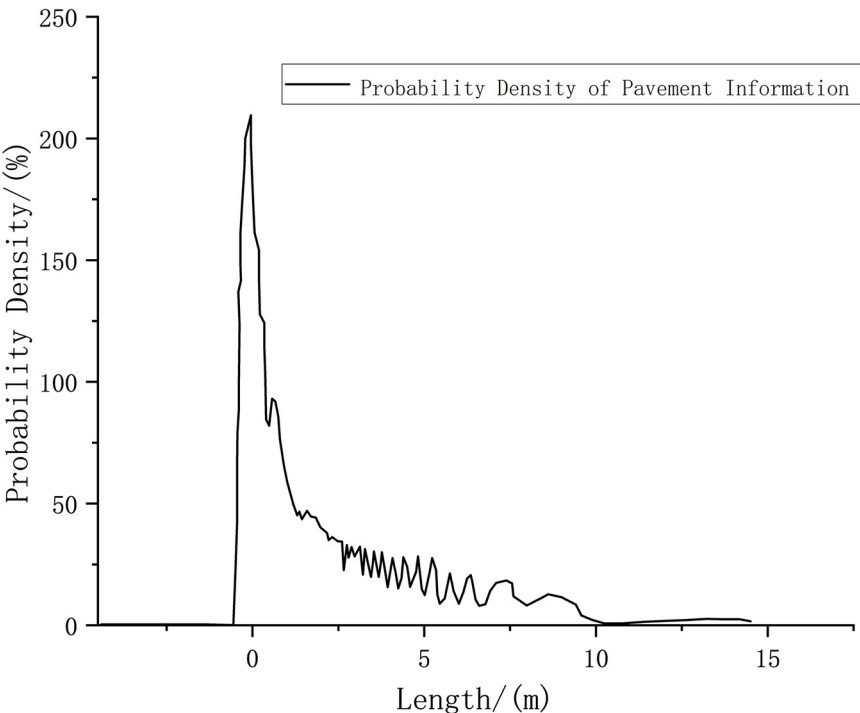

**Fig 14. Road surface probability density after vehicle travel for 9 m.**

distance of 8 cm from the ground was placed in front of the LIDAR laser to perform tests for algorithm verification. The test site is shown in Fig 8.

The horizontal distance from the sensor to the center of the arch obstacle was set to 10 m as the starting point. The obstacle information measured at this time is shown in Fig 9.

Figs 9 and 10 show that the probability density value of pavement elevation data can be obtained by conducting the tests within the LIDAR detection range of 0–10 m. Beyond 10 m,

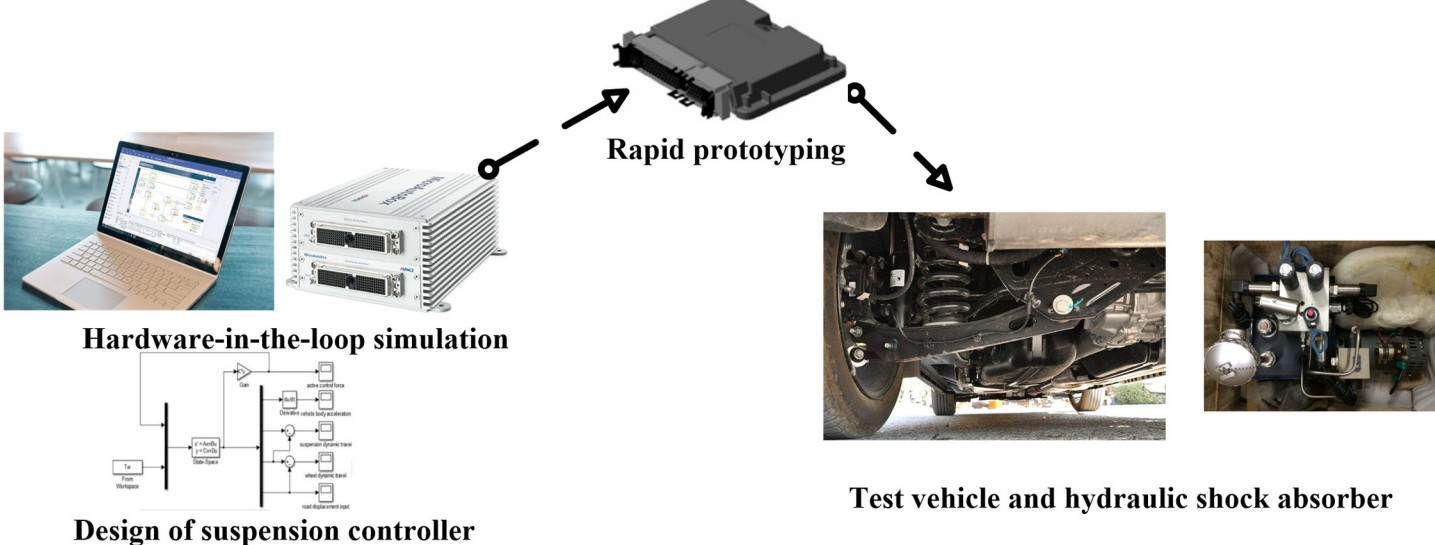

**Fig 15. Test vehicle and hydraulic shock absorber.**

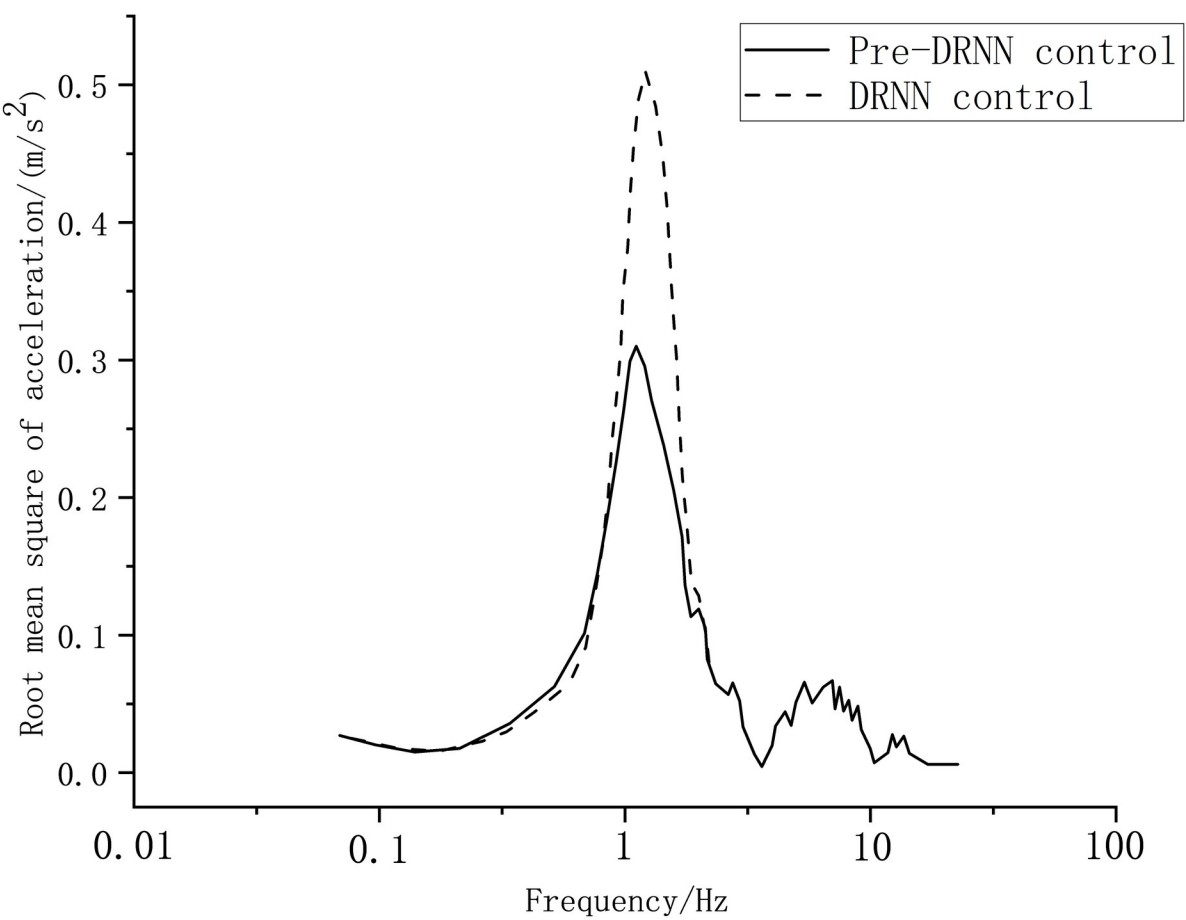

**Fig 16. Suspension vertical celebration.**

the peak value of the measured probability density tends to be close to 0, and the probability distribution is not concentrated; therefore, measurement accuracy cannot be achieved. Within the detection range of 10 m, the obtained pavement elevation fluctuates by approximately –0.05 m. The measured road information contains noise that is caused by the more common fluctuations and jumps in the sensor data.

Figs 11 and 12 depict that once the vehicle has covered a distance of 5 m, after many iterations of the recursive algorithm, the pavement elevation contour can be partially displayed. Furthermore, a peak value appears at 5 m, which indicates that the accuracy of the measured data has been improved at this time. Because the adjoint recursive algorithm superimposes the LIDAR data, the data information density is improved.

Figs 13 and 14 show that, after the car has traveled 9 m, LIDAR is able to obtain an obstacle contour that is closer to the real contour by increasing the number of iterations of the algorithm. Each position of the arch used to simulate pavement elevation obtains a large probability density, and the noise is reduced.

In summary, the low density of the LIDAR point cloud data affects the accuracy of the pavement elevation detection, and the detection accuracy cannot be guaranteed owing to factors such as the LIDAR data jump. Through the LIDAR continuous scanning recursive matching algorithm, the obtained ground elevation gradually approaches the actual contour with an increase in scanning times and algorithm iteration times.

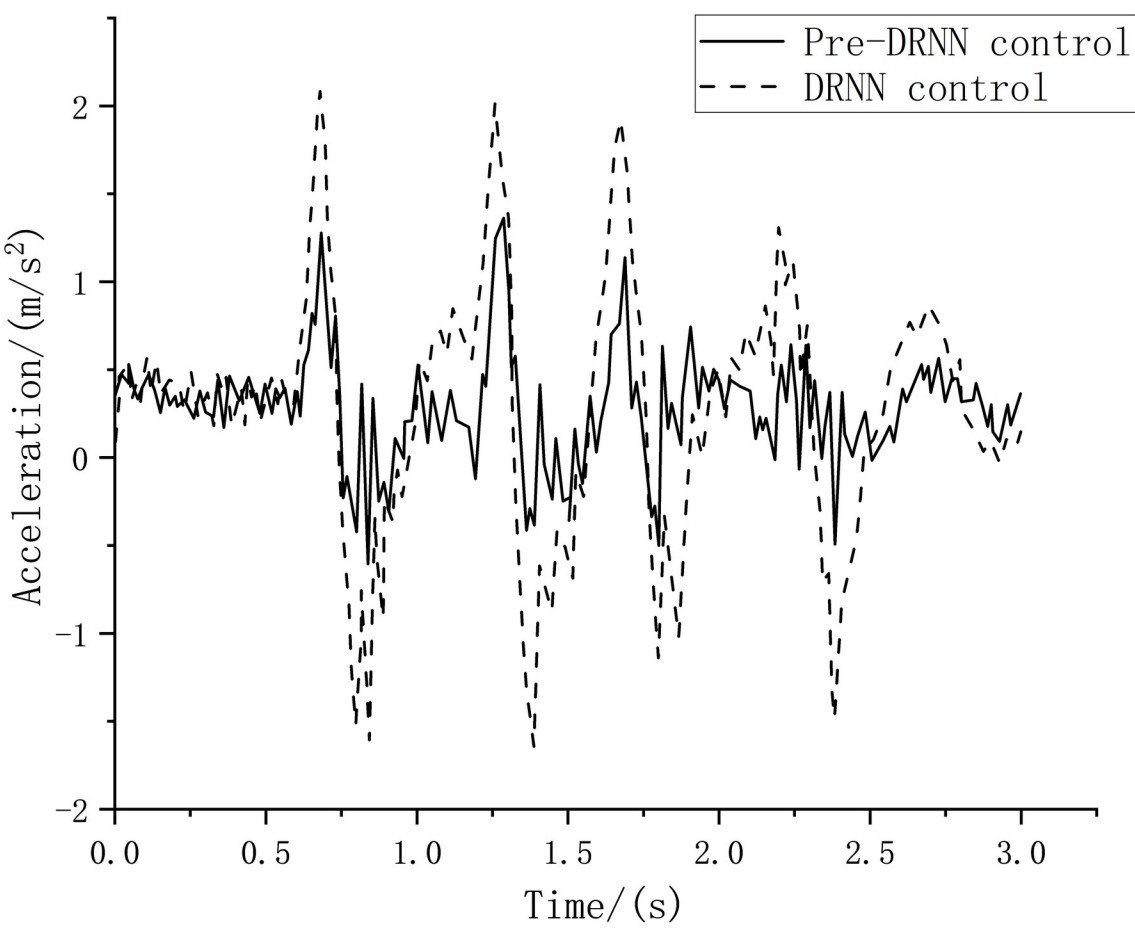

**Fig 17. Left front seat acceleration.**

## 5.2 Vehicle test and verification

In this study, a special vehicle with a semi-active suspension actuator was considered as the research object, and a hardware-in-the-loop co-simulation was conducted based on the Simulink control model and dSPACE real-time simulation system. Furthermore, the automatically generated source code was written into the controller for the suspension test. The test vehicle and hydraulic shock absorber are shown in Fig 15. The active vehicle suspension system was analyzed according to the diagonal recursive neural network controller designed in the previous section, and the vehicle speed was set to 20 m/s.

Generally, the root mean square of acceleration over a period of time can be used to evaluate the suspension vibration. The test vehicle was tested to ascertain the vibration reduction on bumpy roads, and it can be seen from Fig 16 that the root mean square value of the vertical acceleration of the suspension system reached 3.2416 m/s$^2$ in 0–4 s when the DRNN control algorithm was adopted for semi-active suspension control. However, once the road elevation preview algorithm was integrated into the system, the root mean square value of the vertical acceleration was 2.7312 m/s$^2$ with the pre-DRNN strategy control. The damping performance of the semi-active suspension improved by 15.75%. Moreover, the driving comfort was improved.

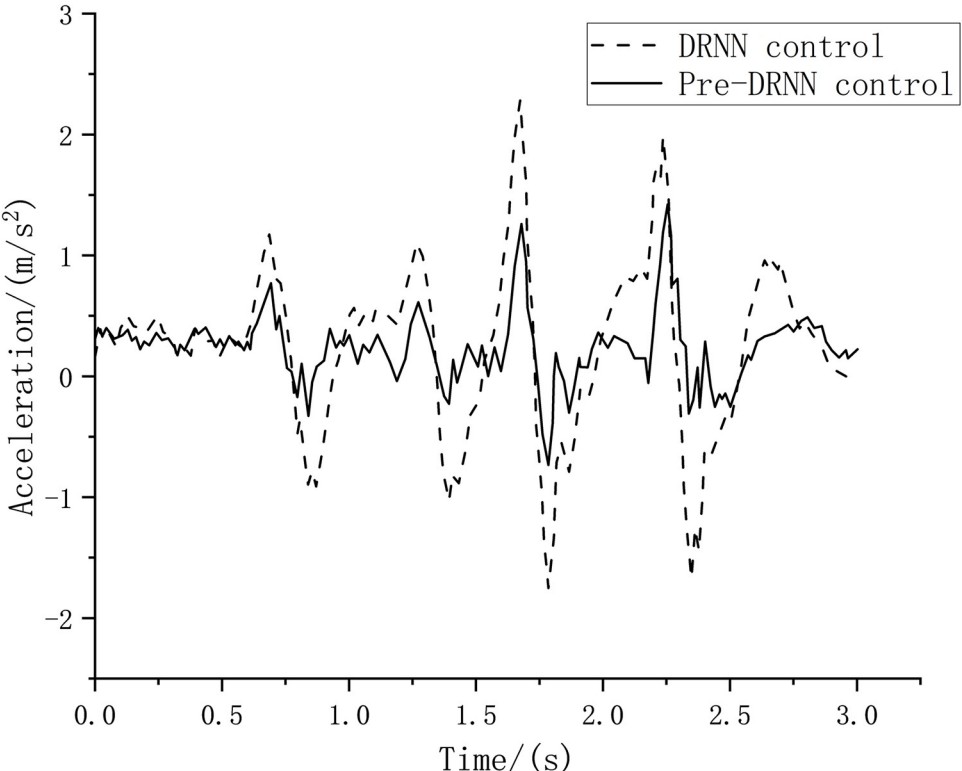

**Fig 18. Right rear seat acceleration.**

In terms of tire dynamic travel, only the left front wheel was considered as an example. Under bumpy conditions, the peak dynamic stroke of the left-front tire with the DRNN control method was 3.4 mm, while the peak dynamic stroke with the pre-DRNN control algorithm was 2.4 mm. These results highlight the pre-DRNN control effect.

This result can also be verified using the vibration analysis of the vehicle seats. Taking the left front seat and the right rear seat as an example, as shown in Figs 17 and 18, when the intermittent bumpy road excitation is applied, the root mean square value of the seat acceleration resulting from the Pre-DRNN algorithm is lower than that of the DRNN algorithm. This indicates that the control method can effectively improve passenger comfort.

## 6. Conclusion

In this paper, a recursive matching algorithm based on the continuous scanning of a single-line LIDAR sensor is proposed to obtain accurate foresight road elevation information. Consequently, the adjustment parameters of the control quantity of the semi-active suspension actuator can be determined. The semi-active suspension controller adopts the pre-DRNN algorithm. A simulation platform and a real vehicle test platform were built; after parameter debugging and verification, the obtained results revealed that the proposed method can effectively control the semi-active suspension and improve ride comfort and stability.

## Supporting information

**S1 Appendix. The experimental test results.**
(RAR)

## Author Contributions

**Data curation:** Zhengcai Yang, Chuan Shi, Yinglin Zheng, Shirui Gu.

**Formal analysis:** Chuan Shi, Yinglin Zheng.

**Validation:** Shirui Gu.

**Writing – original draft:** Shirui Gu.

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
