## [Decision Letter · Decision Letter 0]

22 Mar 2022

PONE-D-22-05626A Study on a Vehicle Active Suspension Control System Based on Road Elevation IdentificationPLOS ONE

Dear Dr. zheng-cai,

Thank you for submitting your manuscript to PLOS ONE. After careful consideration, we feel that it has merit but does not fully meet PLOS ONE’s publication criteria as it currently stands. Therefore, we invite you to submit a revised version of the manuscript that addresses the points raised during the review process.

We look forward to receiving your revised manuscript.

Kind regards,

Feng Chen

Academic Editor

PLOS ONE

Journal Requirements:

Reviewers' comments:

Reviewer's Responses to Questions

**Comments to the Author**

1. Is the manuscript technically sound, and do the data support the conclusions?

Reviewer #1: Yes

Reviewer #2: Yes

2. Has the statistical analysis been performed appropriately and rigorously? 

Reviewer #1: Yes

Reviewer #2: Yes

3. Have the authors made all data underlying the findings in their manuscript fully available?

Reviewer #1: Yes

Reviewer #2: Yes

4. Is the manuscript presented in an intelligible fashion and written in standard English?

Reviewer #1: Yes

Reviewer #2: No

5. Review Comments to the Author

Reviewer #1: This paper proposes a vehicle active suspension control system based on road elevation identification, which seems quite interesting and promising.

There are some small areas that need to be improved.

1. The literature review of this paper lacks a sense of hierarchy. The review of pavement recognition and control strategies should be clearly described.

2. It is suggested to point out the problems more clearly and explain how the proposed method solves these problems.

3. The content in Figure 1 is different from the text description. Please optimize the Figure 1 to correctly express Pavement elevation recognition method.

4. In the Section IV System Overall Scheme Design, ∆h is used to represent the difference between the road roughness at the preview point and the road roughness at the sub-preview point. It is not strict to judge road shape only by ∆h transient value at a certain time. The authors are suggested to further revise it.

Reviewer #2: 1. Figure 2 does not clearly describe the physical meaning of nL, i.e., ‘relative pitch angle between the vehicle body and wheel’.

2. In Figure 2, X0-Y0-Z0 is the vehicle body fixed coordinate system?

3. LIDAR is a keyword of the paper. It is better to define this term at the beginning of the paper. Thus, the term of ‘laser LIDAR’ appeared in the context may be simplified as ‘LIDAR’.

4. In Equations (3) and (4), ‘z’ is used, but in the sentence below Equation (3), ‘Z’ is utilized. The principle of consistency needs to be followed.

5. Below Equation (1), nL is defined as ‘Relative pitch angle between the vehicle body and wheel’, while below Equation (5), nL is redefined as ‘Disturbances caused by vertical bumps and other movements of the vehicle’. For a single symbol, double definitions need to be avoided.

6. It is not clear if vector components are introduced in Equations (6) and (7). Based on the second equation of equation set (7), it seems that all components on the equation should be scalars instead of vectors. If this is the case, the arrow signs need to be removed in order to avoid confusion.

7. On page 8, it is stated: “… and the probability density of the measurement point can be approximated by a continuous Gaussian normal distribution function…”. It is not clear what is the base for this assumption.

8. The vehicle body fixed coordinate system shown in Figure 4 should be consistent with that illustrated in Figure 2.

9. As shown in Figure 4, a seven DOF vehicle model is generated in the study. It is not clear if the model is validated using experimental data or other methods.

10. It seems that only shock absorber damping is controlled during the operation of the suspension. If this is the case, the term of semi-active suspension should be used instead of active suspension.

Typos:

1) In the third paragraph on page 3: “The first category are …” � The first category is …

2) In the same paragraph: “The second category are …” � The second category is …

Improvement of the English of the paper:

For example:

1) In the last paragraph on page 5, it is stated: “The continuous scanning recursive matching algorithm proposed in this study improves the accuracy of the contour elevation of the obstacle in front of the vehicle by performing multiple continuous scanning recursive matching on the laser pulse echo signal.” This sentence may be improved by using the word ‘intends’ and minor change: “The continuous scanning recursive matching algorithm proposed in this study intends to improve the accuracy of …”

2) On page 6, “According to the actual scanning angle of the vehicle-mounted laser LIDAR, the measuring point of each scan was approximately 0o to 45o from the horizontal position to the road.” This sentence may be rewritten as: “As shown in Figure 2, the scanning angle of the vehicle-mounted LIDAR may vary from 0o to 45o, and at the two extreme angles the measuring point on the ground surface is located at infinite long distance and at the nearest detectable point on the surface.”

3) On page 15, “Assuming that the vehicle body is rigid, the suspension motion has three degrees of freedom of vertical vibration, pitch, and roll, …” This sentence needs to be rewritten.

6. PLOS authors have the option to publish the peer review history of their article (what does this mean?). If published, this will include your full peer review and any attached files.

Reviewer #1: No

Reviewer #2: No

---

## [Author Response · Author response to Decision Letter 0]

30 Apr 2022

Response to Reviewer 1 Comments

Dear Reviewers:

Thank you for your letter and for the reviewers' comments concerning our manuscript entitled "A Study on a Vehicle Semi-Active Suspension Control System Based on Road Elevation Identification" （ID:PONE-D-22-05626）.Those comments are all valuable and very helpful for revising and improving our paper, as well as the important guiding significance to our researches. We have studied comments carefully and made correction which we hope meet with approval. Revised portion are marked in yellow in the paper. The main corrections in the paper and the responds to the reviewer's comments are as flowing:

Point 1: The literature review of this paper lacks a sense of hierarchy. The review of pavement recognition and control strategies should be clearly described. 

Response 1: It is really true as Reviewer suggested that the literature review of this paper is lack of a sense of hierarchy,and we have revised this part according to the Reviewer's suggestion. In the introduction, we introduce and analyze more literature technical methods, and categorize and summarize the characteristics of these methods.

Point 2: It is suggested to point out the problems more clearly and explain how the proposed method solves these problems.

Response 2: After analyzing the characteristics of the supplemented literature by category, the main problems existing in the previous research are summarized, and the solution of this paper is proposed.Please see "lines 111 to 120" in the revised document.

Point 3: The content in Figure 1 is different from the text description. Please optimize the Figure 1 to correctly express Pavement elevation recognition method.

Response 3: We have improved the logic shown in Figure 1, which is mainly reflected in the refinement of the data transmission process in the road height recognition algorithm.

Point 4: In the Section IV System Overall Scheme Design, ∆h is used to represent the difference between the road roughness at the preview point and the road roughness at the sub-preview point. It is not strict to judge road shape only by ∆h transient value at a certain time. The authors are suggested to further revise it.

Response 4: Just depending on whether the difference between the road roughness of the preview point and the road roughness of the sub-preview point is greater than zero is indeed not enough to accurately represent the change rate of different road roughness. According to international standards, we re established the corresponding rules between road roughness difference and road grade, and quantified the evaluation standard of roughness change rate. Please see "lines 418 to 427" in the revised document and the new figure 6.

Response to Reviewer 2 Comments

Dear Reviewers:

Thank you for your letter and for the reviewers' comments concerning our manuscript entitled "A Study on a Vehicle Semi-Active Suspension Control System Based on Road Elevation Identification" （ID:PONE-D-22-05626）.Those comments are all valuable and very helpful for revising and improving our paper, as well as the important guiding significance to our researches. We have studied comments carefully and made correction which we hope meet with approval. Revised portion are marked in yellow in the paper. The main corrections in the paper and the responds to the reviewer's comments are as flowing:

Point 1: Figure 2 does not clearly describe the physical meaning of nL, i.e., ‘relative pitch angle between the vehicle body and wheel’. 

Response 1: We have improved Figure 2, aligned it with the vehicle coordinate system in Figure 4, and showed the pitching action of the vehicle body relative to the wheel. In the figure, the pitching angle nL is marked.

Point 2: In Figure 2, X0-Y0-Z0 is the vehicle body fixed coordinate system?

Response 2: Since we have modified Figure 2 and aligned it with the vehicle coordinate system in Figure 4, X0-Y0-Z0 in Figure 2 becomes the vehicle coordinate system with the vehicle centroid as the origin.

Point 3: LIDAR is a keyword of the paper. It is better to define this term at the beginning of the paper. Thus, the term of ‘laser LIDAR’ appeared in the context may be simplified as ‘LIDAR’.

Response 3: We have defined lidar at the beginning. Please check 'line 128 to line 130' in the revised document, and replace all the term of‘laser LIDAR’appeared in the context involved in the text with "LIDAR".

Point 4: In Equations (3) and (4), ‘z’ is used, but in the sentence below Equation (3), ‘Z’ is utilized. The principle of consistency needs to be followed.

Response 4: We have replaced 'Z' in the sentence below Equation (3) with 'z', please see 'Line 163' in the revised document.

Point 5: Below Equation (1), nL is defined as ‘Relative pitch angle between the vehicle body and wheel’, while below Equation (5), nL is redefined as ‘Disturbances caused by vertical bumps and other movements of the vehicle’. For a single symbol, double definitions need to be avoided.

Response 5: We have replaced the symbol "disturbance caused by vehicle vertical turbulence and other motion" under equation (5) with "pL" instead of "nL". Please see "line 168" in the revised document.

Point 6: It is not clear if vector components are introduced in Equations (6) and (7). Based on the second equation of equation set (7), it seems that all components on the equation should be scalars instead of vectors. If this is the case, the arrow signs need to be removed in order to avoid confusion.

Response 6: All components in equations (6) and (7) are changed from vector to scalar, and remove all potentially confusing the arrow signs in the paper.

Point 7: On page 8, it is stated: “… and the probability density of the measurement point can be approximated by a continuous Gaussian normal distribution function…”. It is not clear what is the base for this assumption.

Response 7:It is really true as Reviewer suggested that the paper is not clear what is the base for this assumption,and we have re-written this part according to the Reviewer's suggestion. Based on the uneven distribution of pulse signals emitted by lidar, we have supplemented the probability density assumption basis of measurement points in this paper. Please see 'lines 192 to 196' in the revised document.

Point 8: The vehicle body fixed coordinate system shown in Figure 4 should be consistent with that illustrated in Figure 2.

Response 8: We have adjusted the vehicle coordinate system in Figure 2 to make it consistent with the vehicle coordinate system in Figure 4, so the body coordinate system in Figure 4 has not been modified.

Point 9: As shown in Figure 4, a seven DOF vehicle model is generated in the study. It is not clear if the model is validated using experimental data or other methods.

Response 9: The seven DOF vehicle model established in this study can reflect the actual motion state of the vehicle. It is a classic model in the industry. The geometric parameters, mechanical parameters (stiffness and damping) and mass parameters (mass and moment of inertia) of the model are derived from the real vehicle data.

Point 10: It seems that only shock absorber damping is controlled during the operation of the suspension. If this is the case, the term of semi-active suspension should be used instead of active suspension.

Response 10: We have replaced all the term of active suspension in this paper with semi-active suspension.

We appreciate for Reviewers warm work earnestly, and hope that the correction will meet with approval.

Once again, thank you very much for your comments and suggestions.

---

## [Decision Letter · Decision Letter 1]

20 May 2022

A Study on a Vehicle Semi-Active Suspension Control System Based on Road Elevation Identification

PONE-D-22-05626R1

Dear Dr. zheng-cai,

We’re pleased to inform you that your manuscript has been judged scientifically suitable for publication and will be formally accepted for publication once it meets all outstanding technical requirements.

Kind regards,

Feng Chen

Academic Editor

PLOS ONE

Additional Editor Comments (optional):

Reviewers' comments:

Reviewer's Responses to Questions

**Comments to the Author**

1. If the authors have adequately addressed your comments raised in a previous round of review and you feel that this manuscript is now acceptable for publication, you may indicate that here to bypass the “Comments to the Author” section, enter your conflict of interest statement in the “Confidential to Editor” section, and submit your "Accept" recommendation.

Reviewer #1: (No Response)

Reviewer #2: All comments have been addressed

2. Is the manuscript technically sound, and do the data support the conclusions?

Reviewer #1: (No Response)

Reviewer #2: Yes

3. Has the statistical analysis been performed appropriately and rigorously? 

Reviewer #1: (No Response)

Reviewer #2: Yes

4. Have the authors made all data underlying the findings in their manuscript fully available?

Reviewer #1: (No Response)

Reviewer #2: Yes

5. Is the manuscript presented in an intelligible fashion and written in standard English?

Reviewer #1: (No Response)

Reviewer #2: Yes

6. Review Comments to the Author

Reviewer #1: (No Response)

Reviewer #2: The revised version of the paper addresses almost all of the reviewer's comments/concerns. Therefore, the reviewer recommends that the revised version be accepted for publication in the journal.

7. PLOS authors have the option to publish the peer review history of their article (what does this mean?). If published, this will include your full peer review and any attached files.

Reviewer #1: No

Reviewer #2: No

---

## [Editor Report · Acceptance letter]

31 May 2022

PONE-D-22-05626R1 

A Study on a Vehicle Semi-Active Suspension Control System Based on Road Elevation Identification 

Dear Dr. zheng-cai:

I'm pleased to inform you that your manuscript has been deemed suitable for publication in PLOS ONE. Congratulations! Your manuscript is now with our production department. 

Kind regards, 

on behalf of

Dr. Feng Chen 

Academic Editor

PLOS ONE